# The impact of increasing expenditure on National Essential Public Health Services on the medical costs of hypertension in China: A difference-in-difference analysis

**Long Xue**[1]*, **Mengyun Sui**[1], **YunZhen He**[1], **Hongzheng Li**[1], **Xiaohua Ying**[1,2]*

**1** School of Public Health, Fudan University, Shanghai, China, **2** NHC Key Laboratory of Health Technology Assessment, Fudan University, Shanghai, China

* 18111020030@qq.com (LX); xhying@fudan.edu.cn (XY)

## Abstract

### Background

The prevention and control of hypertension should be an effective way to reduce deaths and it has been a high priority in China. In 2013, the Chinese government increased the subsidy standard for the National Essential Public Health Services Package (NEPHSP) from RMB 15 to RMB 30 per person, which was expected to cover 70 million hypertensions. This study explored the influence of increasing NEPHSP subsidy on outpatient and inpatient expenditure among patients with hypertension.

### Methods

Data were mined from the 2011–2015 Harmonized China Health and Retirement Longitudinal Study. The study sample included 3192 hypertensive patients who were not lost to follow-up from 2011 to 2015. Hypertensive patients who covered by NEPHSP from 2011 to 2015 were defined as the treatment group, otherwise defined as the comparison group. The policy intervention was the increase of NEPHSP subsidy in 2013, and the years before and after 2013 were respectively considered as pre- (2011) and post-intervention (2015). The primary outcomes variables were the outpatient and inpatient expenditure of patients with hypertension, based on direct spending of outpatients and inpatients separately reported by patients with hypertension. Using propensity score matching (PSM) to match the individual characteristics of hypertension in the treatment group and the comparison group, difference-in-differences (DID) were used to analyze the outcomes.

### Results

The patients with hypertension' outpatient and inpatient expenditure patterns in the treatment and control group show an increasing trend from 2011 to 2015. After PSM, of the 1 956 hypertensive participants, 369 covered by the NEPHSP before and after 2013. A DID estimate of the increased NEPHSP subsidy was associated with a significant decrease of 1 251.35 RMB (t = 2.13, P = 0.034) in hypertension related inpatient expenditure, no significant change (t = 0.61, P = 0.544) among outpatient expenditure.

**Data Availability Statement:** All the data we used have been publicly released on the CHARLS website: https://charls.charlsdata.com/pages/data/111/zh-cn.html.

**Funding:** The authors received no specific funding for this work.

**Competing interests:** The authors have declared that no competing interests exist.

## Conclusions

The NEPHSP may reduce inpatient expenditure among hypertension. Further strengthening of the NEPHSP may reduce their burden.

## Introduction

Two recent studies of hypertension in China suggest that the prevalence of hypertension in China was 23.2% among people aged ≥18 and over from 2012 to 2015 [1], and 44.7% among people aged 35–75 from 2014 to 2017 [2]. Moreover, hypertension increased from 18% [3] to 23.2% [1] from 2002 to 2015. At the same time, several studies using the criteria of the Chinese guidelines for the management of hypertension (systolic BP (SBP) ≥140 mmHg, diastolic BP (DBP) ≥90 mmHg) [4] indicate that less than 50% of patients were awareness, 30–40% were taking antihypertensive medications, and 7.2–15.3% had control [1,2,5]. Using the same nationwide survey but applying the 2017 American College of Cardiology/American Heart Association Guidelines for high blood pressure, hypertension prevalence in China reached 46.4%, while the blood pressure control rate dropped to 3.0% [1].

Cardio-cerebrovascular diseases caused by untreated and inadequately treated hypertension have been the main cause of death among Chinese people [6]. As prevention and control of hypertension should be an effective way to reduce deaths [7], it has been a high priority in China [8]. To address current health challenges, including obesity, hypertension, and non-communicable diseases, and to reduce the disparities for the Chinese population in accessing essential public health services, the Chinese government issued the National Essential Public Health Services Package (NEPHSP) in 2009 [9]; this initiative provides free public health services to meet the challenges posed by hypertension. It includes physical examination, health education, regular health checkups, and regular follow-ups provided to patients with hypertension aged ≥ 35 [10] (see S1 Table for details). At inception, citizens received public health subsidies of RMB 15 from the government, which increased to RMB 30 in 2013, and the government increased support for patients with hypertension from 45 million in 2011 to 70 million in 2013 [11].

Previous studies have indicated that after increasing the subsidy of NEPHSP, patients with hypertension covered by the NEPHSP was associated with an increase of 7.9%, 10.3%, and 10.5% in hypertension the rate of control, medication, and monitoring [9]. Since increasing the subsidy of NEPHSP, the treatment rate and control rate of hypertensive patient increased, so can the outpatient and hospitalization expense of hypertensive patient rise? So far, no relevant research has been conducted. Further, previous studies on the cost of hypertension treatment did not include the effect of the increasing the subsidy of NEPHSP [12–14].

In this study, we used nationally representative longitudinal data to analyze 1) the changes and differences in outpatient and inpatient expenses of hypertensive patients before and after 2013 the increasing the subsidy of NEPHSP and 2) the impact of the increasing the subsidy of NEPHSP on outpatient and hospitalization costs of hypertensive patients.

## Materials and methods

### Data sources

Data were obtained from the China Health and Retirement Longitudinal Study (CHARLS), a nationally representative longitudinal survey of individuals aged ≥ 45. The national baseline

survey was conducted in 2011, with wave 2 in 2013, wave 3 in 2015. It aims to measure the health status, economic status, and well-being of Chinese residents. The survey includes a rich set of questions regarding economic standing, physical and psychological health, demographics, and social networks of aged persons. The population included in the baseline survey was asked the same core questions every two years.

The data distributed across 450 villages in 150 counties [15]. The CHARLS used a multistage probability sampling approach to select a nationally representative sample. Specifically, the first stage involved random sampling, using the probability-proportional-to-size method, that included all county-level units of China with the exception of Tibet, with the final sample comprising 150 countries. The sample was stratified by region and within region by urban or rural status. In the second stage, administrative villages in rural areas and neighborhoods in urban areas were randomly selected as primary sampling units (PSUs). Three PSUs were selected from each county. In the third stage, 24 households were randomly selected based on the geographical locations and lists of each PSU. In the fourth and final stage, a resident aged ≥45 years was randomly selected from a household, and an interview was conducted with both the selected resident and their spouse. For cross-national comparisons of other international aging surveys, a Harmonized CHARLS was created to coordinate the CHARLS with the Health and Retirement Survey in the United States, for which detailed information is publicly available online [16].

## Study population

Patients with hypertension were defined according to "A doctor has told you that you have hypertension," and hypertension criteria was based on the 2010 Chinese guidelines for the management of hypertension: systolic BP (SBP) ≥140 mm Hg, diastolic BP (DBP) ≥90 mm Hg [4].

The treatment group was covered by NEPHSP in 2011 and 2015, and the comparison group were not covered by NEPHSP in 2011 and 2015. To determine whether hypertensive patients were covered by NEPHSP, we use the same method as ZHANG et al. [9], using the question "When did you have your last physical examination?" to identify responders who had received physical exams. The question "Who paid for your last physical examination?" measured whether the medical costs were covered by the NEPHSP, given that the NEPHSP includes a free physical exam at least once a year for patients with hypertension. Respondents who chose "government" were the treatment group, and respondents who chose "non-government" had not received a free medical exam and were defined as the comparison group.

## Interventions

The policy intervention was the increase of NEPHSP subsidy in 2013, so the year before 2013 (2011) was defined as pre-intervention, and the year after 2013 (2015) was defined as post-intervention.

## Outcome measurement

This study reported spending of outpatients and inpatients separately and used the data in in pre-and post-intervention as the dependent variable in the models. All outpatient expenditure in the past month was recorded, including both treatment and medication costs. Inpatient expenditure was defined as all the inpatient costs during the past year, including fees paid to the hospital, ward fees but excluding wages paid to a hired nurse, transportation costs, and accommodation costs for yourself or family members. To identify total hospitalization expenditure in the past year, total medical cost of doctor visits, and amount paid by their insurance

company, we used the following items from the CHARLS baseline questionnaire: "How many times have you received inpatient care?" "How many times did you visit a medical facility?" "What is your total hospitalization cost?" and "What is your total outpatient cost?" If the respondent had two or more inpatient or outpatient treatments in the past year or month, then the respondent was asked to list the total medical costs for all visits.

## Covariates

The covariate variables for this study consisted of individual socioeconomic characteristics and health information, such as age (45–54, 55–64 and $\geq$ 65), sex (Men or Women), marital status (Cohabitant or Single), occupation (Farmer or not), education was grouped according to the 1997 International Classification of Standards in Education (1 = Less than lower secondary education, 2 = Upper secondary & vocational training, and 3 = Tertiary education). Household income was classified by four groups according to the 2011 household income quartile: poor (< RMB 2,320), low income (RMB 2,320–11,200), middle income (RMB 11,200–32,400), and high income ($\geq$ RMB 32,400). Other characteristics were health insurance (have or not), Body mass index (BMI) according to the WHO recommendation for the Asian and South Asian population, and is coded as follows: 0 = Normal weight (BMI 18.5–23kg/cm$^2$); 1 = Underweight (BMI < 18.5 kg/cm$^2$), 2 = Overweight or obesity (BMI $\geq$23 kg/cm$^2$) [17], and self-reported health status (0 = Fair; 1 = Poor; 2 = Good; 3 = Very good or Excellent).

## Statistical analysis

As indicated above, respondents covered by NEPHSP in 2011and 2015 were defined as the treatment group, while the comparison group had not received this service. First, the covariates of the two groups of hypertensive patients in 2011 were matched by propensity score matching (PSM) [18], which enables us to calculate weights based on the socioeconomic characteristics and health information of patients with hypertension that yield unbiased estimates of the impact of factors of interest [19]. Patients' age, sex, marital, work, education, house income, insurance, residence, BMI, health state was matching variables. Logistic regression on each eligible subject was matched according to the propensity score, and because the sample sizes were big difference between the treatment and comparison groups (584 vs. 2 608), using the nearest-neighbor method at a 1:4 ratio with 0.02 in the caliper [20]. The 50 times bootstrap method was employed to obtain the robust standard error. We retained the matched data for the final analysis.

Second, based on the matched data, a difference-in-differences (DID) method was used to analyze the changes in outpatient and inpatient expense for hypertensive patients in pre-and post-intervention. DID can effectively detect the intervention effect between the treatment and comparison groups and isolate the time trend unrelated to the intervention [21]. The effect of the NEPHSP was estimated by comparing the differences between two changes in outcomes: (1) changes between pre- and post- intervention within the treatment group and (2) the pre- and post-intervention periods in the comparison group [22]. This estimate was computed from a regress model, which includes two dichotomous variables: the time (pre- or post- intervention), the group (treatment or comparison group) and an interaction term between the time and the group. The impact was estimated through the coefficient of this interaction term. The model allows adjustment for potential confounders (age, sex, marital, work, education, house income, insurance, residence, BMI, health state) to reduce residual confounders [23]. Given that the skewed distribution of outpatient and inpatient expense violated the normal distribution assumption of the regress model, logarithmic transformation was adopted in the analysis [24].

Data analyses were performed using Stata v.16.0 (StataCorp, Texas, USA). Chi-square test was used to test the difference between the categorical data. A P-value <0.05 was considered statistically significant.

### Ethics approval and informed consent statement

Ethical approval for collecting data on human subjects was obtained from the Biomedical Ethics Review Committee of Peking University (IRB00001052-11015). All participants provided written informed consent. All participants provided written informed consent.

## Results

### Sample characteristics

Before PSM, a total of 3 192 hypertensive patients who were not lost to follow-up were included in the study, including 584 patients in the treatment group and 2608 patients in the comparison group. After matching, there were 369 cases in the treatment group and 1587 cases in the comparison group (Fig 1).

Table 1 presents the socioeconomic and health characteristics (N = 3,192) in 2011. Compared with treatment group, the age ($\chi2$ = 24.50, $P<0.001$) and health status ($\chi2$ = 8.63, $P$ = 0.035) in the comparison group was significantly different. There were no significant differences in sex, marital, education, household income, health insurance, residence, and BMI between the two groups.

### Baseline characteristics after PSM

Table 2 illustrates the socioeconomic and health characteristics before and after the PSM in 2011. Before matching, the treatment group (n = 2 608) and comparison group (n = 548) differed significantly in terms of age $\geq$60 (t = 4.61, p< 0.001) and self-reported poor health (t = -2.26, p = 0.025). After matching, the differences in the socioeconomic and health characteristics between the two groups were insignificant. 215 participants in the treatment group were lost after matching, which may indicate selection bias. For this reason, we compared the matched sample with the original sample, and no significant differences were found between the two groups, as shown in S2 Table.

S1–S4 Figs illustrate the standardized bias across the covariates in the -10% to 10% range, propensity score distribution, and kernel density across the two groups' propensity scores before and after matching, and the matching effect was satisfactory.

### Outpatient and inpatient expenditure changes pre- and post-NEPHSP

Figs 2 and 3 presents before and after PSM, the patients with hypertension' outpatient and inpatient expenditure patterns in the treatment group and the control group show an increasing trend from 2011 to 2015. After PSM, there was a difference in hospitalization costs between the treatment group and the control group (2888.91 VS. 1837.49, t = 1.73, P = 0.084) in 2015.

### Common trend assumption

Fig 4 illustrates the expenditure of outpatients and inpatients, compared between the comparison and treatment groups in pre-and post-intervention, respectively, after PSM. No statistically significant differences for the outpatient (p = 0.473) and inpatient expenses (p = 0.730) in the pre-intervention period estimates were observed.

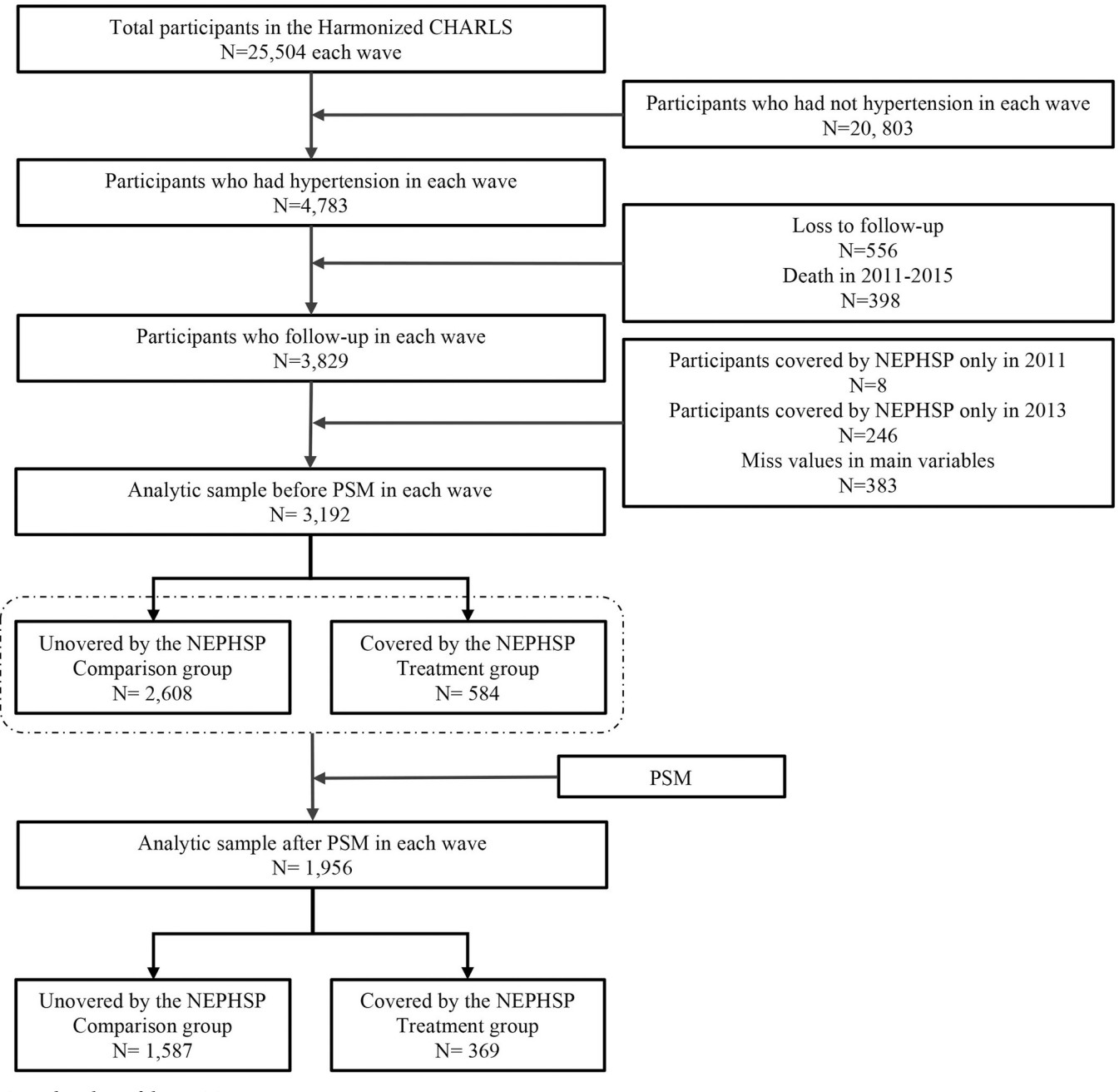

**Fig 1. Flow chart of the participants.**

## Comparing the treatment group and the comparison group

Table 3 presents the results from the DID analyses of outpatient and inpatient expenditure changes between treatment and comparison groups after PSM. Compared with pre-intervention, the treatment group was associated with a decrease of 266.23 RMB in outpatient expenditure post-intervention, but there was no statistically significant (t = 0.61, P = 0.544). In terms of inpatient expenditure, the treatment group was associated with a significant decrease of 1251.35 RMB (t = 2.13, P = 0.034) in inpatient expenditure post-intervention compared with pre-intervention.

**Table 1. Baseline characteristics of patients with hypertension (% 95%CI).**

| | Comparison (%, 95%CI) | | | Treatment (%, 95%CI) | | | χ2 | p |
|---|---|---|---|---|---|---|---|---|
| N | 2,608 | | | 584 | | | | |
| **Age** | | | | | | | | |
| 45–54 | 19.70 | 16.11 | 23.87 | 28.18 | 26.64 | 29.78 | 32.32 | <0.001 |
| 55–64 | 37.68 | 33.09 | 42.52 | 40.56 | 38.86 | 42.29 | | |
| ≥65 | 42.61 | 37.87 | 47.49 | 31.25 | 29.66 | 32.89 | | |
| **Sex** | | | | | | | | |
| Men | 46.32 | 41.52 | 51.20 | 43.86 | 42.14 | 45.59 | 1.19 | 0.276 |
| Women | 53.68 | 48.80 | 58.48 | 56.14 | 54.41 | 57.86 | | |
| **Marital** | | | | | | | | |
| Cohabitant | 80.88 | 76.76 | 84.42 | 80.58 | 79.17 | 81.92 | 0.01 | 0.905 |
| Single | 19.12 | 15.58 | 23.24 | 19.42 | 18.08 | 20.83 | | |
| **Work** | | | | | | | | |
| Not farmer | 25.25 | 21.26 | 29.70 | 23.33 | 21.89 | 24.84 | 1.00 | 0.317 |
| Farmer | 74.75 | 70.30 | 78.74 | 76.67 | 75.16 | 78.11 | | |
| **Education** | | | | | | | | |
| Less than lower secondary education | 91.91 | 88.83 | 94.20 | 89.55 | 88.43 | 90.57 | 4.19 | 0.123 |
| Upper secondary & vocational training | 7.11 | 4.98 | 10.05 | 8.81 | 7.87 | 9.85 | | |
| Tertiary education | 0.98 | 0.37 | 2.59 | 1.64 | 1.25 | 2.15 | | |
| **Household income** | | | | | | | | |
| Poor | 25.53 | 20.76 | 30.97 | 25.04 | 23.23 | 26.93 | 3.73 | 0.292 |
| Low income | 24.47 | 19.78 | 29.85 | 24.99 | 23.19 | 26.88 | | |
| Middle income | 22.34 | 17.84 | 27.60 | 25.41 | 23.60 | 27.32 | | |
| High income | 27.66 | 22.73 | 33.20 | 24.56 | 22.77 | 26.45 | | |
| **Health insurance** | | | | | | | | |
| No | 5.64 | 3.77 | 8.35 | 6.64 | 5.82 | 7.56 | 0.94 | 0.331 |
| Yes | 94.36 | 91.65 | 96.23 | 93.36 | 92.44 | 94.18 | | |
| **Residence** | | | | | | | | |
| Urban | 40.20 | 35.53 | 45.05 | 39.60 | 37.91 | 41.31 | 0.08 | 0.779 |
| Rural | 59.80 | 54.95 | 64.47 | 60.40 | 58.69 | 62.09 | | |
| **BMI (kg/cm2)** | | | | | | | | |
| Normal weight (18.5–23) | 30.86 | 26.14 | 36.02 | 28.09 | 26.37 | 29.87 | 2.38 | 0.305 |
| Underweight (<18.5) | 4.15 | 2.47 | 6.90 | 3.63 | 2.97 | 4.43 | | |
| Overweight or obesity (≥23) | 64.99 | 59.72 | 69.91 | 68.28 | 66.44 | 70.07 | | |
| **Self-reported health status** | | | | | | | | |
| Fair | 50.49 | 44.88 | 56.10 | 46.63 | 44.58 | 48.69 | 8.63 | 0.035 |
| Poor | 29.84 | 24.95 | 35.23 | 35.21 | 33.27 | 37.20 | | |
| Good | 12.46 | 9.19 | 16.68 | 12.91 | 11.59 | 14.36 | | |
| Very good or Excellent | 7.21 | 4.79 | 10.73 | 5.24 | 4.40 | 6.24 | | |

BMI: Body mass index.

## Discussion

Our results revealed that the inpatient expenditure of patients with hypertension decreased after NEPHSP's subsidy was increased. Previous research indicates that increasing NEPHSP's subsidy improved the rate of hypertension control, antihypertensive medication use, and blood pressure monitoring [9]. Other research has indicated that providing free public health

**Table 2. Comparable characteristics between the two groups after PSM.**

| | Before matching | | | | After matching | | | |
|---|---|---|---|---|---|---|---|---|
| | Comparison | Treatment | *t* | *p* | Comparison | Treatment | *t* | *p* |
| **N** | 2608 | 584 | | | 1587 | 369 | | |
| **Age** | | | | | | | | |
| 45–54 | Ref. | | | | | | | |
| 55–64 | 37.69 | 40.56 | -1.11 | 0.266 | 31.71 | 34.15 | -0.47 | 0.640 |
| ≥65 | 42.61 | 31.25 | 4.61 | 0.000 | 54.27 | 52.74 | 0.28 | 0.783 |
| **Sex** | | | | | | | | |
| Men | Ref. | | | | | | | |
| Women | 53.68 | 56.14 | -0.94 | 0.345 | 42.68 | 38.57 | 0.76 | 0.449 |
| **Marital** | | | | | | | | |
| Cohabitant | Ref. | | | | | | | |
| Single | 19.12 | 19.42 | -0.14 | 0.885 | 21.95 | 16.62 | 1.22 | 0.222 |
| **Work** | | | | | | | | |
| Not farmer | Ref. | | | | | | | |
| Farmer | 74.76 | 76.67 | -0.86 | 0.392 | 78.05 | 78.35 | -0.07 | 0.947 |
| **Education** | | | | | | | | |
| Less than lower secondary education | Ref. | | | | | | | |
| Upper secondary & vocational training | 7.11 | 8.81 | -1.15 | 0.249 | 6.10 | 4.57 | 0.61 | 0.541 |
| Tertiary education | 0.98 | 1.64 | -1.01 | 0.311 | 0.61 | 1.22 | -0.58 | 0.563 |
| **Household income** | | | | | | | | |
| Poor | Ref. | | | | | | | |
| Low income | 24.47 | 24.99 | -0.19 | 0.850 | 31.10 | 31.10 | 0.00 | 1.000 |
| Middle income | 22.34 | 25.42 | -1.12 | 0.263 | 18.29 | 18.14 | 0.04 | 0.972 |
| High income | 27.66 | 24.56 | 1.13 | 0.259 | 24.39 | 23.78 | 0.13 | 0.898 |
| **Health insurance** | | | | | | | | |
| No | Ref. | | | | | | | |
| Yes | 94.36 | 93.36 | 0.77 | 0.441 | 95.73 | 96.80 | -0.51 | 0.612 |
| **Residence** | | | | | | | | |
| Urban | Ref. | | | | | | | |
| Rural | 59.80 | 40.56 | -1.11 | 0.266 | 67.68 | 72.26 | -0.90 | 0.368 |
| **BMI (kg/cm2)** | | | | | | | | |
| Normal weight (18.5–23) | Ref. | | | | | | | |
| Underweight (<18.5) | 4.15 | 3.63 | 0.48 | 0.631 | 6.71 | 4.42 | 0.90 | 0.368 |
| Overweight or obesity (≥23) | 64.99 | 68.28 | -1.22 | 0.223 | 59.76 | 64.33 | -0.85 | 0.395 |
| **Self-reported health status** | | | | | | | | |
| Fair | Ref. | | | | | | | |
| Poor | 29.84 | 35.21 | -2.26 | 0.025 | 32.32 | 30.03 | 0.45 | 0.656 |
| Good | 12.46 | 14.91 | -1.85 | 0.064 | 12.81 | 10.37 | 0.69 | 0.492 |
| Very good or Excellent | 7.21 | 5.25 | 1.42 | 0.156 | 3.05 | 2.44 | 0.34 | 0.736 |

BMI: Body mass index.

services to patients with hypertension can effectively reduce their burden [25]. The NEPHSP may have a positive influence that is related to improvements in hypertension diagnosis, treatment, and control. Indeed, some signs point to a shift toward reactive care for prevention [26], which may be positively correlated with promoting healthy lifestyle traits, such as a low sodium diet [27], physical activity, weight loss, and smoking and drinking cessation [28]. It

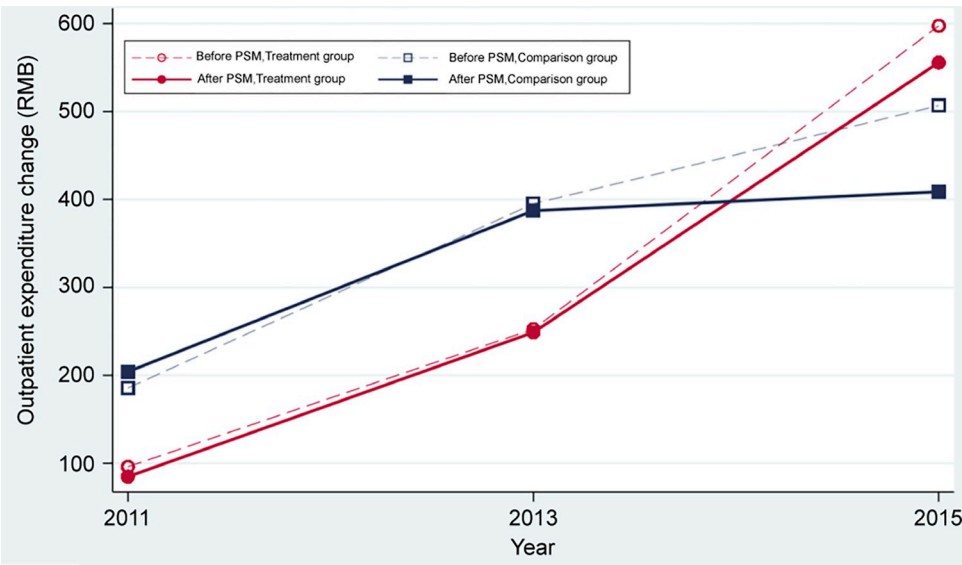

**Fig 2. Comparison of outpatient expenditure of the treatment and comparison groups before and after PSM (RMB).**

may also signal decreases in hospitalization rates, lengths of stay, or procedures for complications [29]. The NEPHSP provides free preventive care services for patients with hypertension to prevent or delay comorbidity and complications of the disease and lower overall healthcare costs. These results are similar to those of a study conducted in Japan that demonstrated that free screening and treatment services for patients with hypertension were associated with a 42–75% reduction in stroke incidence [30]. Thus, it is not surprising that in China, the expanded NEPHSP can reduce the disease burden of patients with hypertension.

There was an overall upward trend in expenditure associated with hypertension from 2011 to 2015, which might be related to the increasing incidence of hypertension in China. China is

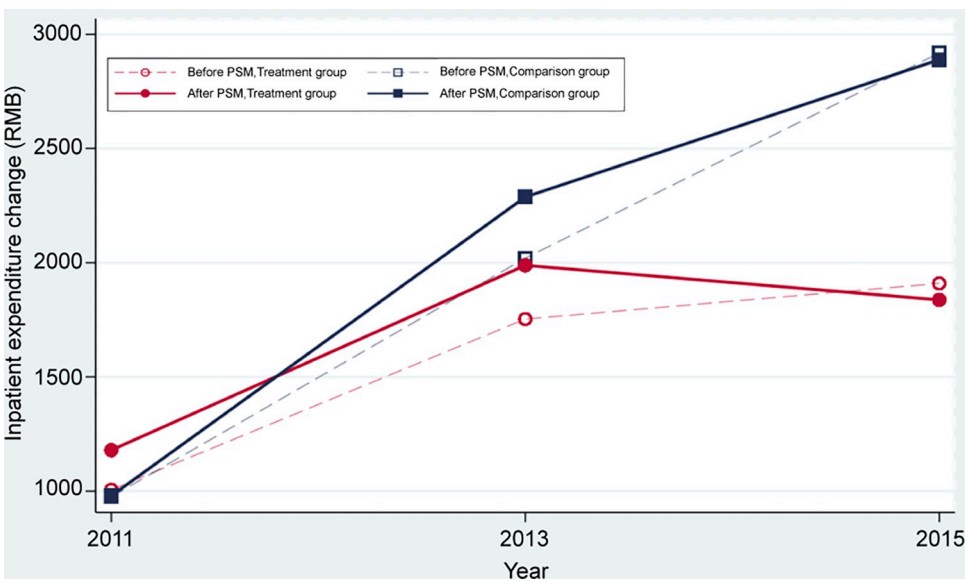

**Fig 3. Comparison of inpatient expenditure of treatment and control comparison before and after PSM (RMB).**

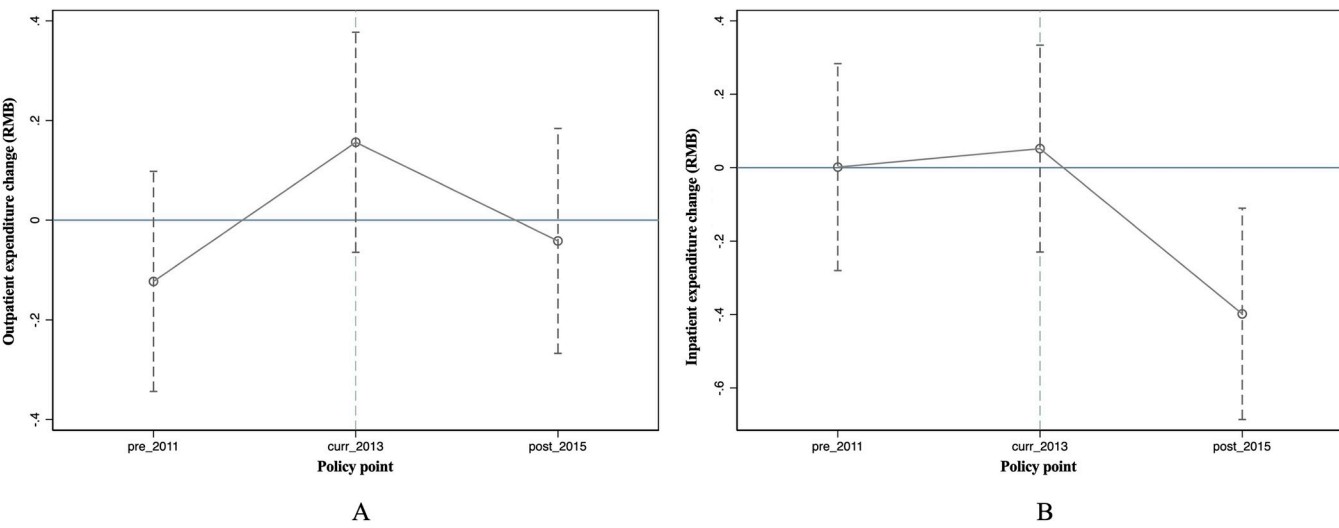

**Fig 4.** A. Trends in outpatient expenditure after PSM (RMB). B. Trends in inpatient expenditure after PSM (RMB).

one of the fastest growing countries all over the world [31], the prevalence of self-reported hypertension among people aged 15 and over rose from 6.7% in 2008 to 14.2% in 2013. Meanwhile, the number of patients with hypertension older than 65 increased from 21.6% to 37.0% and the number of patients with hypertension aged 45 to 64 also doubled during this period [32]. The life expectancy of patients with hypertension and incidence of young hypertensive patients increased. Correspondingly, healthcare expenditure increased, as people with hypertension live longer and their chances of developing comorbidities and complications increase [28,33]. One study in China revealed that 87% of heart disease deaths, 71% of stroke deaths, 54% of ischemic heart disease deaths, 41% of other cardiovascular disease deaths, and 43% of chronic kidney disease deaths were attributable to hypertension [34], which may also be one of the reasons for the increased cost for patients with hypertension observed in this study.

After 2013, Those covered by the NEPHSP displayed an upward tendency in outpatient services and a downward tendency in hospitalization services. These findings suggest that the expand of the NEPHSP might be achieving its goals of improving hypertension management. The same results have been confirmed in other studies in China [9,35]. Compared with 2002, the number of patients who were aware they had hypertension increased by 54 million, the

**Table 3. Impact of the NEPHSP on the outpatient and inpatient expenditure after PSM.**

| | Comparison ($\bar{x}$ (sd)) | Treatment ($\bar{x}$ (sd)) | Marginal difference |
|---|---|---|---|
| **Inpatient** | | | |
| pre-intervention | 979.61(-5840.3) | 1179.54(-5247.45) | 199.93(590.23) |
| post-intervention | 2888.91(-14059.57) | 1837.49(-12068.14) | -1051.42*(607.67) |
| DID estimates | | | -1251.35**(859.37) |
| **Outpatient** | | | |
| pre-intervention | 204.13(-1411.47) | 84.89(-426.71) | -119.24(166.41) |
| post-intervention | 408.74(-2726.68) | 555.73(-6030.02) | 146.99(117.67) |
| DID estimates | | | 266.23(203.81) |

*p<0.10

**p<0.05.

number who were treated increased by 53 million, and the number whose blood pressure was under control increased by 0.25 million. For people covered by NEPHSP, higher awareness rates lead to more outpatient visits and fewer complications lead to lower hospitalization costs. This is largely because of the government's continued efforts to reform the healthcare system and equalize public health services.

We noted that healthcare costs for patients with high blood pressure increased more slowly in the comparison groups after 2013. This may be due to "spillover effects" from the NEPHSP. The same spillover effects have been observed in other policy studies [36,37]. Our results suggest that there might be a positive spillover effect in comparison group.

We acknowledge that there are several limitations to this study. First, although PSM was used to eliminate some individual selection bias factors, there are numerous confounding factors, such as the living environment, social network, access to health care in the place of residence, medical consumer price level, and healthcare supply in the place of residence. Second, our dependent variable fails to isolate the costs purely because of hypertension, and other comorbidities of the patients with hypertension may interfere with the results. Future studies must link the survey data and the medical treatment data to obtain the pure cost of hypertension. Moreover, there was a considerable amount of zero values in the dependent variable, which may affect the result estimates, as we cannot determine whether the cost was due to patient visits or non-visits to healthcare providers. Third, outpatient and inpatient costs were based on respondents' recollections and may have been missed and overreported, adversely affecting our estimates. In the future, we will use the medical bills to increase the accuracy of the cost information. Fourth, Indirect measurement of intervention in this study would affect the results of the article, and we need to find cleaner interference indicators in the future. Finally, we used data for patients with hypertension aged 45 and over in the CHARLS, but patients with hypertension aged 35 and over were covered by the NEPHSP, which may have influenced our estimates. In the future, we will use more nationally representative samples and include more variables to study this issue.

## Conclusion

In China, the expand of the NEPHSP may have reduced outpatient and inpatient treatment expense for patients with hypertension, although outpatient costs were not significant. The NEPHSP is an important health policy that establishes a file for hypertensive patients, regular check-ups and encourages patients to seek medical care reasonably and regularly. In the future, financial compensation should be increased, the catalog of free screening services for chronic diseases should be expanded, the occurrence of chronic disease complications should be avoided or delayed, and the burden of chronic diseases should be reduced. For policymakers, the NEPHSP may have begun to achieve its goals of increasing the health awareness of patients with hypertension and reducing their financial burden.

## Supporting information

**S1 Fig. Test of standardized percentage bias across covariates before and after propensity score matching.** Illustrates the standardized bias across the covariates in the -10% to 10% range across the two groups' propensity scores before and after matching, and the matching effect was satisfactory.
(TIF)

**S2 Fig. Test of propensity score distribution before and after propensity score matching.** Illustrates the propensity score distribution across the two groups' propensity scores before

and after matching, and the matching effect was satisfactory.
(TIF)

**S3 Fig. Test of propensity score kernel density before propensity score matching.** Illustrates the propensity kernel density across the two groups' propensity scores before matching.
(TIF)

**S4 Fig. Test of propensity score kernel density after propensity score matching.** Illustrates the propensity kernel density across the two groups' propensity scores after matching, and the matching effect was satisfactory.
(TIF)

**S1 Table. 2009 National Essential Public Health Services Package (NEPHSP) content.** Description of data: The Chinese government issued the National Essential Public Health Services Package (NEPHSP) in 2009; this initiative provides free public health services, including health education, regular health checkups, and regular follow-ups.
(DOCX)

**S2 Table. Comparison of samples before and after PSM.** Description of data: Comparison of samples data before and after matching, and the groups no significant difference.
(DOCX)

## Acknowledgments

We thank the China Health and Retirement Longitudinal Study (CHARLS) team for providing data.

## Author Contributions

**Conceptualization:** Long Xue, Xiaohua Ying.

**Data curation:** Long Xue, Mengyun Sui, YunZhen He, Hongzheng Li.

**Formal analysis:** Long Xue, Mengyun Sui, YunZhen He, Hongzheng Li.

**Investigation:** Mengyun Sui, YunZhen He.

**Methodology:** Long Xue, Xiaohua Ying.

**Supervision:** Xiaohua Ying.

**Writing – original draft:** Long Xue.

**Writing – review & editing:** Long Xue, Xiaohua Ying.

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
