## [Decision Letter · Decision Letter 0]

20 Sep 2022

PONE-D-22-22961The impact of increasing expenditure on National Essential Public Health Services on the medical costs of hypertension in China: A difference-in-difference analysisPLOS ONE

Dear Dr. Long,

Thank you for submitting your manuscript to PLOS ONE. After careful consideration, we feel that it has merit but does not fully meet PLOS ONE’s publication criteria as it currently stands. Therefore, we invite you to submit a revised version of the manuscript that addresses the points raised during the review process.

We look forward to receiving your revised manuscript.

Kind regards,

Jalandhar Pradhan

Academic Editor

PLOS ONE

Journal Requirements:

"NO - Include this sentence at the end of your statement: The funders had no role in study design, data collection and analysis, decision to publish, or preparation of the manuscript"

"We also thank all members of the NHC Key Laboratory of Health Technology Assessment for providing assistance."

"NO - Include this sentence at the end of your statement: The funders had no role in study design, data collection and analysis, decision to publish, or preparation of the manuscript"

Additional Editor Comments:

Authors are requested to submit the revised draft based the comment and suggestions shared with you. The final decision will be taken after having another round of review.

Reviewers' comments:

Reviewer's Responses to Questions

**Comments to the Author**

1. Is the manuscript technically sound, and do the data support the conclusions?

Reviewer #1: Partly

Reviewer #2: Yes

2. Has the statistical analysis been performed appropriately and rigorously? 

Reviewer #1: No

Reviewer #2: Yes

3. Have the authors made all data underlying the findings in their manuscript fully available?

Reviewer #1: Yes

Reviewer #2: Yes

4. Is the manuscript presented in an intelligible fashion and written in standard English?

Reviewer #1: Yes

Reviewer #2: Yes

5. Review Comments to the Author

Reviewer #1: Overall, the writing of this manuscript did not meet the criteria of a scientific paper. The introduction did not describe the significance of this study clearly. The methods are difficult to follow. The intervention was not measured directly. The results did not show the major findings quantitatively.

Abstract:

- The methods should describe the study sample selection and intervention and define primary outcomes.

- The first sentence in the results repeated the last sentence in the methods. Additionally, the results should describe the major findings quantitatively. Using "significant decrease" alone is not sufficient.

Introduction:

- The purpose of this is to examine the effect of NEPHSP on health care costs in patients with hypertension. What is the background of initiating NEPHSP? What are the barriers to hypertension treatment in China?

- Line 28: What is the time period of incidence?

- Lines 36-37: How do Chinese clinical guidelines define hypertension and treatment goals?

- The introduction (lines 51-52) identified some gaps in the literature. Did your study address these gaps? Based on the methods and results, your study did not assess medication use, blood pressure monitoring, or hypertension-related costs. It seems that the introduction did not reflect the significance of your study.

- Line 50: Using "positively associated with..." is too general. Please describe the findings quantitatively.

Methods:

- The description of the data source is not clear. What do you mean the survey was administered every two years since 2011? Do you mean the survey was started in 2011 and followed the participants till 2015? Was the survey disseminated every two years to the same participants, i.e., 2011, 2013, and 2015? Did the survey ask the same questions during each cycle?

- What information was collected by the survey?

- What sampling method was used in this survey? Did this survey use a complex survey design?

- What is the purpose of this survey?

- What are the inclusion and exclusion criteria of the study sample?

- Was self-reported hypertension from the diagnosis of hypertension told by doctors or participants' perceptions?

- Line 71: what do you mean all included participants were not covered by NEPHSP?

- Line 77: How to define outpatient and inpatient spending?

- Lines 79 – 80: Is it difficult to understand "Direct medical costs were included in inpatient expenditure?" What is the definition of medical cost in this study? Usually, medical costs include inpatient and outpatient expenses.

- Line 87: How could patients report the expenses accurately, including the amount paid by insurance and out-of-pocket payment? If the cost information just relied on participants' memory, this would be a limitation.

- Lines 94-97: It seems that you made an assumption to define the intervention indirectly based on the question, "Who paid for your last physical examination?" If so, is it possible that some participants could lose coverage paid by the government during the follow-up period? Why were some participants covered by government insurance and some not? Additionally, lines 70-71 mentioned all participants were not covered by NEPHSP. How to explain the inconsistency? The indirect measurement of intervention to define treatment and control groups is a big concern.

- Line 102: The survey population represents people >=45 years. Why did you use >=65 or not to define age? Shouldn't it be 45-64 and 65+?

- Line 104: The education level is confusing. Do you mean Middle school represents the highest education level?

- Line 109: Usually, the health status is categorized into poor-fair, good, very good-excellent. Did the survey use poor and not poor to define the health status? If so, the category of not poor is too broad and could not show the true association between health status and outcome.

- Line 113: What gap? Do you mean the characteristics of the two groups were significantly different?

- Lines 114-115: How did you estimate propensity score? Did you use logistic regression? What covariates were included? If you used 1:4 matching, there should be a big difference in the sample size between the two groups. What is the original sample size for each group?

- Lines 125-126: How to define pre- and post- NEPHSP within treatment group? All people in the treatment had been covered by NEPHSP since 2011. Where to get the data before 2011?

- The outcome measures are not clear. The detailed difference-in-difference in costs should be elaborated. In the linear regression model, what is the dependent variable?

Results:

- Before propensity score matching, what is the original sample size for each group? When the propensity score matching was applied, did you include all participants in the treatment for matching or lose some participants? According to Figure 1, you lost 215 participants in the treatment group after matching. The concern is whether the matched sample can still represent the original study sample. Is there a selection bias in the study design?

- Table 1: p values are needed to show the significance of all the variables between the two groups. Usually, the first-year characteristics reflect the baseline characteristics. Why did you list characteristics in 2015?

- The results should describe the major findings in Tables ad figures quantitatively. Using "significant difference" is too general.

- Did you compare the results in Table 3 before and after PSM?

Conclusion:

- The conclusion should be based on the results, and the conclusion paragraph seems to be about policy implications.

Reviewer #2: This paper used data from the 2011-2015 Harmonized China Health and Retirement Longitudinal Study to assess the impact of China’s National Essential Public Health Services Package (NEPHSP) on hypertension services utilization and expenditure. The authors used difference-in-differences with propensity score matching to control unobserved heterogeneities. The paper is well-written and with rigor. I have some minor comments for the authors to consider.

First, the manuscript needs a careful copyediting with occasional misuses of English. I will list some specific examples later.

Second, while the methods appear to be well-executed, some clarifications are useful. For instance, why “1:4 matching” is used? What variables are used in defining “neighbor”? Expenditures tend to be skewed, is there any effort to address it with specification tests and generalized linear models?

Specific Comments

Line 140: Table 1: A t-test comparing the “treatment” and the “control” group would be useful.

Line 149: Table 2, “Not poor self-reported health” should be rephrased, maybe “self-reported good or better health”

Line 22-23: it should be “in hypertension related inpatient expenditure”

Line 169-173: need a concise title for Figure 4 – could put some of the information in a footnote.

Line 178: The authors occasionally used “compared to” – note that “compare to” highlights similarities while “compare with” stresses on the differences.

Line 102-108: Sex should be (Men or women), “living with or without a partner” could be cohabitant – might need to be clarified or recoded. Education categories could be cleaned up as well. The income thresholds need to have a brief justification – it will be confusing to readers why those are used. “insurance” – should be health insurance. By the way, what are the insurance?

6. PLOS authors have the option to publish the peer review history of their article (what does this mean?). If published, this will include your full peer review and any attached files.

Reviewer #1: No

Reviewer #2: No

---

## [Author Response · Author response to Decision Letter 0]

11 Oct 2022

The author made a comprehensive format modification according to the requirements of the magazine

"NO - Include this sentence at the end of your statement: The funders had no role in study design, data collection and analysis, decision to publish, or preparation of the manuscript"

The authors have added this sentence at the fund of statement for the article that the funders had no role in study design, data collection and analysis, decision to publish, or preparation of the manuscript, and the same change have been made in cover letter.

"We also thank all members of the NHC Key Laboratory of Health Technology Assessment for providing assistance."

"NO - Include this sentence at the end of your statement: The funders had no role in study design, data collection and analysis, decision to publish, or preparation of the manuscript"

"We also thank all members of the NHC Key Laboratory of Health Technology Assessment for providing assistance." This sentence has been removed from the article in the Acknowledgments Section.

The authors added a full ethical statement and informed consent in the "Methods and methods" section of the article.

Abstract:

- The methods should describe the study sample selection and intervention and define primary outcomes.

1. Study sample selection: 

The study sample included 3192 hypertensive patients who were not lost to follow-up from 2011 to 2015.

2. Intervention: 

The policy intervention was the increase of NEPHSP subsidy in 2013, and the years before and after 2013 were respectively considered as pre-intervention (2011) and post-intervention (2015).

3. Define primary outcomes

The primary outcomes variables were the outpatient and inpatient expenditure of patients with hypertension, based on direct spending of outpatients and inpatients separately reported by patients with hypertension.

- The first sentence in the results repeated the last sentence in the methods. Additionally, the results should describe the major findings quantitatively. Using "significant decrease" alone is not sufficient.

1. The first sentence in the results repeated the last sentence in the methods.

The author has deleted the repeated sentence from the result section：

“Using propensity score matching (PSM) to match the individual characteristics of hypertension in the NEPHSP-covered group and the NEPHSP-uncovered group, difference-in-differences (DID) were used to analyze the outcomes.”

2. the results should describe the major findings quantitatively. Using "significant decrease" alone is not sufficient.

According to the reviewers, the authors have added a quantitative description of the main findings：

A DID estimate of the increased NEPHSP subsidy was associated with a significant decrease of 1251.35 RMB (t=2.13, P=0.034) in hypertension related inpatient expenditure, no significant change (t=0.61, P=0.544) among outpatient expenditure.

Introduction:

- The purpose of this is to examine the effect of NEPHSP on health care costs in patients with hypertension. What is the background of initiating NEPHSP? What are the barriers to hypertension treatment in China?

1. What is the background of initiating NEPHSP?

As prevention and control of hypertension should be an effective way to reduce deaths, it has been a high priority in China. To address current health challenges, including obesity, hypertension, and non-communicable diseases, and to reduce the disparities for the Chinese population in accessing essential public health services, the Chinese government issued the National Essential Public Health Services Package (NEPHSP) in 2009 

2. What are the barriers to hypertension treatment in China?

I think the main barriers to hypertension treatment in China is low awareness and treatment rates：

Several studies using the criteria of the Chinese guidelines for the management of hypertension (systolic BP (SBP) ≥140 mmHg, diastolic BP (DBP) ≥90 mmHg) indicate that less than 50% of patients were aware of their condition, 30–40% were taking prescribed antihypertensive medications, and 7.2–15.3% had achieved control . Using the same nationwide survey but applying the 2017 American College of Cardiology/American Heart Association Guidelines for high blood pressure, hypertension prevalence in China reached 46.4%, while the blood pressure control rate dropped to 3.0% .

- Line 28: What is the time period of incidence?

We increased the time period for prevalence studies：

Two recent studies of hypertension in China suggest that the prevalence of hypertension in China was 23.2% among people aged ≥18 and over from 2012 to 2015, and 44.7% among people aged 35–75 from 2014 to 2017

- Lines 36-37: How do Chinese clinical guidelines define hypertension and treatment goals?

We have added a description of the Chinese guidelines for the management of hypertension. 

“At the same time, several studies using the criteria of the Chinese guidelines for the management of hypertension (systolic BP (SBP) ≥140 mmHg, diastolic BP (DBP) ≥90 mmHg) ”

- The introduction (lines 51-52) identified some gaps in the literature. Did your study address these gaps? Based on the methods and results, your study did not assess medication use, blood pressure monitoring, or hypertension-related costs. It seems that the introduction did not reflect the significance of your study.

This is a very good suggestion, and we will revise this section of the introduction based on this suggestion：

Previous studies have indicated that after increasing the subsidy of NEPHSP, patients with hypertension covered by the NEPHSP was associated with an increase of 7.9%, 10.3%, and 10.5% in the rate control, medication, and monitoring . Since increasing the subsidy of NEPHSP, the treatment rate and control rate of hypertensive patient increased, so can the outpatient and hospitalization expense of hypertensive patient rise? So far, no relevant research has been conducted.

- Line 50: Using "positively associated with..." is too general. Please describe the findings quantitatively.

we will revise this section based on reviewer suggestion：

Previous studies have indicated that after increasing the subsidy of NEPHSP, patients with hypertension covered by the NEPHSP was associated with an increase of 7.9%, 10.3%, and 10.5% in hypertension the rate of control, medication, and monitoring 

Methods:

- The description of the data source is not clear. What do you mean the survey was administered every two years since 2011? Do you mean the survey was started in 2011 and followed the participants till 2015? Was the survey disseminated every two years to the same participants, i.e., 2011, 2013, and 2015? Did the survey ask the same questions during each cycle?

Data were obtained from the China Health and Retirement Longitudinal Study (CHARLS), a nationally representative longitudinal survey of individuals aged ≥ 45. The national baseline survey was conducted in 2011, with wave 2 in 2013, wave 3 in 2015. 

The population included in the baseline survey was asked the same core questions every two years.

- What information was collected by the survey?

The survey includes a rich set of questions regarding economic standing, physical and psychological health, demographics, and social networks of aged persons. 

- What sampling method was used in this survey? Did this survey use a complex survey design?

The CHARLS used a multistage probability sampling approach to select a nationally representative sample. Specifically, the first stage involved random sampling, using the probability-proportional-to-size method, that included all county-level units of China with the exception of Tibet, with the final sample comprising 150 countries. The sample was stratified by region and within region by urban or rural status. In the second stage, administrative villages in rural areas and neighborhoods in urban areas were randomly selected as primary sampling units (PSUs). Three PSUs were selected from each county. In the third stage, 24 households were randomly selected based on the geographical locations and lists of each PSU. In the fourth and final stage, a resident aged ≥45 years was randomly selected from a household, and an interview was conducted with both the selected resident and their spouse . 

- What is the purpose of this survey?

CHARLS aims to measure the health status, economic status, and well-being of Chinese residents aged ≥45 years.

- What are the inclusion and exclusion criteria of the study sample?

Patients with hypertension were included in the study. 

Patients with hypertension were defined according to "A doctor has told you that you have hypertension," and hypertension was defined according to 2010 Chinese guidelines for the management of hypertension: systolic BP (SBP) ≥140 mmHg, diastolic BP (DBP) ≥90 mm Hg 

- Was self-reported hypertension from the diagnosis of hypertension told by doctors or participants' perceptions?

Patients with hypertension were defined according to "A doctor has told you that you have hypertension." 

- Line 71: what do you mean all included participants were not covered by NEPHSP?

The treatment group was covered by NEPHSP in 2011 and 2015, and the comparison group were not covered by NEPHSP in 2011 and 2015.

- Line 77: How to define outpatient and inpatient spending?

All outpatient expenditure in the past month was recorded, including both treatment and medication costs. 

Inpatient expenditure was defined as all the inpatient costs during the past year, including fees paid to the hospital, ward fees but excluding wages paid to a hired nurse, transportation costs, and accommodation costs for yourself or family members.

To identify total hospitalization expenditure in the past year, total medical cost of doctor visits, and amount paid by their insurance company, we used the following items from the CHARLS baseline questionnaire: “How many times have you received inpatient care?” “How many times did you visit a medical facility?” “What is your total hospitalization cost?” and “What is your total outpatient cost?” If the respondent had two or more inpatient or outpatient treatments in the past year or month, then the respondent was asked to list the total medical costs for all visits.

- Lines 79 – 80: Is it difficult to understand "Direct medical costs were included in inpatient expenditure?" What is the definition of medical cost in this study? Usually, medical costs include inpatient and outpatient expenses.

As the reviewer said, this sentence is easily ambiguous, so the authors remove it and corrected the definitions of inpatient expenses：

Inpatient expenditure was defined as all the inpatient costs during the past year, including fees paid to the hospital, ward fees but excluding wages paid to a hired nurse, transportation costs, and accommodation costs for yourself or family members.

- Line 87: How could patients report the expenses accurately, including the amount paid by insurance and out-of-pocket payment? If the cost information just relied on participants' memory, this would be a limitation.

As the reviewer stated, the fee is based on the respondent's recollection, and we add limitations as stated below：

Third, outpatient and inpatient costs were based on respondents' recollections and may have been missed and overreported, adversely affecting our estimates. In the future, we will use the medical bills to increase the accuracy of the cost information.

- Lines 94-97: It seems that you made an assumption to define the intervention indirectly based on the question, "Who paid for your last physical examination?" If so, is it possible that some participants could lose coverage paid by the government during the follow-up period? Why were some participants covered by government insurance and some not? Additionally, lines 70-71 mentioned all participants were not covered by NEPHSP. How to explain the inconsistency? The indirect measurement of intervention to define treatment and control groups is a big concern.

The authors gave a good suggestion.

we did exclude 246 hypertensive patients who were only covered by NEPHSP in 2011 and 8 patients who were only covered by NEPHSP in 2013, based on which we challenged Figure 1.

Additionally, lines 70-71 mentioned all participants were not covered by NEPHSP. How to explain the inconsistency?

Thank the reviewer for pointing out the slip of the pen in the article.

We initially included people with hypertension, if they were covered by NEPHSP in 2011-2015, which enrolled into the treatment group, otherwise enrolled into the comparison group.

The indirect measurement of intervention to define treatment and control groups is a big concern.

That's a problem, but there's plenty of reason to believe this measurement:

1. As mentioned in the S1 table, the NEPHSP policy includes physical examinations paid for by the government.

2. "Who Paid for the physical examination?" in the questionnaire. The answer was 1 self, 2 relatives, 3 units, 4 government, 5 insurances, 6 etc., which distinguishes government payments from other payments.

3. A previous study by Zhang et al . published in the American Journal of Hypertension used the same method.

4. In the fourth limitation, Indirect measurement of intervention in this study would affect the results of the article, and we need to find cleaner interference indicators in the future.

- Line 102: The survey population represents people >=45 years. Why did you use >=65 or not to define age? Shouldn't it be 45-64 and 65+?

As suggested by reviewers, we reanalyzed the data：

The author has been revised to: 45-54, 55-64 and ≥ 65, and reanalyzed the data.

- Line 104: The education level is confusing. Do you mean Middle school represents the highest education level?

As suggested by reviewers, we reanalyzed the data：

Education variables use a harmonized scale that is a simplified version of the 1997 International Standard Classification of Education (ISCED) codes (www.uis.unesco.org): 1. Less than lower secondary education, 2. Upper secondary & vocational training, and 3. Tertiary education.

- Line 109: Usually, the health status is categorized into poor-fair, good, very good-excellent. Did the survey use poor and not poor to define the health status? If so, the category of not poor is too broad and could not show the true association between health status and outcome.

Based on the reviewers' recommendations：

we classified the health status as 0. Fair; 1. Poor; 2. Good; 3. Very good or Excellent, and reanalyzed the data.

- Line 113: What gap? Do you mean the characteristics of the two groups were significantly different?

Yes, the characteristics of the two groups were significantly different. 

- Lines 114-115: How did you estimate propensity score? Did you use logistic regression? What covariates were included? If you used 1:4 matching, there should be a big difference in the sample size between the two groups. What is the original sample size for each group?

As suggested by reviewers, we added：

Patients’ age, sex, marital, work, education, house income, insurance, residence, BMI, health state was matching variables. Logistic regression on patients’ demographics information for each eligible subject was matched according to the propensity score, and because the sample sizes were big difference between the treatment and comparison groups (408 VS. 3167), using the nearest-neighbor method at a 1:4 ratio with 0.02 in the caliper . The 50 times bootstrap method was employed to obtain the robust standard error.

- Lines 125-126: How to define pre- and post- NEPHSP within treatment group? All people in the treatment had been covered by NEPHSP since 2011. Where to get the data before 2011?

Based on respondent received subsidies of RMB 15 from the government, which increased to RMB 30 in 2013, and the government increased support for patients with hypertension from 45 million in 2011 to 70 million in 2013. The pre- and post- intervention within treatment group was defined as before 2013 (2011) and after 2013 (2015), respectively. 

We focus on whether the outpatient and hospitalization costs of hypertensive patients covered by NEPHSP changed before and after 2013 when the government increased the subsidy of NEPHSP, without considering the use of data before 2011.

- The outcome measures are not clear. The detailed difference-in-difference in costs should be elaborated. In the linear regression model, what is the dependent variable?

As suggested by reviewers, we revised：

1. A difference-in-differences (DID) method was used to analyze the changes in outpatient and inpatient expense for hypertensive patients before and after NEPHSP subsidy increase.

2. This estimate was computed from a regress model, which includes two dichotomous variables: the time (pre- or post- intervention), the group (treatment or comparison group) and an interaction term between the time and the group. The impact was estimated through the coefficient of this interaction term. The model allows adjustment for potential confounders (age, sex, marital, work, education, house income, insurance, residence, BMI, health state) to reduce residual confounders.

3. Given that the skewed distribution of outpatient and inpatient expense data violated the normal distribution assumption of the ordinary linear model, logarithmic transformation was adopted in the analysis.

Results:

- Before propensity score matching, what is the original sample size for each group? When the propensity score matching was applied, did you include all participants in the treatment for matching or lose some participants? According to Figure 1, you lost 215 participants in the treatment group after matching. The concern is whether the matched sample can still represent the original study sample. Is there a selection bias in the study design?

1. Before propensity score matching, what is the original sample size for each group? When the propensity score matching was applied, did you include all participants in the treatment for matching or lose some participants?

Before PSM, a total of 3 192 hypertensive patients who were not lost to follow-up were included in the study, including 584 patients in the treatment group and 2608 patients in the comparison group. After matching, there were 369 cases in the treatment group and 1587 cases in the comparison group (Fig 1).

2. According to Figure 1, you lost 215 participants in the treatment group after matching. The concern is whether the matched sample can still represent the original study sample. Is there a selection bias in the study design?

First, because the sample sizes were big difference between the treatment and comparison groups (584 vs. 2 608), using the nearest-neighbor method at a 1:4 ratio with 0.02 in the caliper.

Second, we tried 1:1,1:2, 1:3,1:4, and 1:5 matching, we determined the optimal matching 1:4. 

Third, we compared the unmatched and matched groups, and no significant differences were found between the two groups, as shown in S2 Table of the Supplementary materials.

- Table 1: p values are needed to show the significance of all the variables between the two groups. Usually, the first-year characteristics reflect the baseline characteristics. Why did you list characteristics in 2015?

It has been modified according to the comments of reviewers. The 2015 variable is deleted, the percentage and 95%CI of each variable in 2011 was displayed, and the chi-square value and P-value were added.

- The results should describe the major findings in Tables and figures quantitatively. Using "significant difference" is too general.

We've stated the chi-square or t values and p-values in all the "significant difference".

- Did you compare the results in Table 3 before and after PSM?

Table 3 presents the results from the DID analyses of outpatient and inpatient expenditure changes between treatment and comparison groups after PSM. Compared to before 2013, the treatment group was associated with a decrease in outpatient expenditure after 2013, but there was no statistically significant (t=0.61, P=0.544). In terms of inpatient expenditure, after 2013, the treatment group was associated with a significant decrease in inpatient expenditure compared with before 2013 (t=2.13, P=0.034).

Conclusion:

- The conclusion should be based on the results, and the conclusion paragraph seems to be about policy implications.

In China, the expand of the NEPHSP may have reduced outpatient and inpatient treatment expense for patients with hypertension, although outpatient costs were not significant. The NEPHSP is an important health policy that establishes a file for hypertensive patients, regular check-ups and encourages patients to seek medical care reasonably and regularly. In the future, financial compensation should be increased, the catalog of free screening services for chronic diseases should be expanded, the occurrence of chronic disease complications should be avoided or delayed, and the burden of chronic diseases should be reduced. For policymakers, the NEPHSP may have begun to achieve its goals of increasing the health awareness of patients with hypertension and reducing their financial burden.

First, the manuscript needs a careful copyediting with occasional misuses of English. I will list some specific examples later.

Second, while the methods appear to be well-executed, some clarifications are useful. For instance, why “1:4 matching” is used? What variables are used in defining “neighbor”? Expenditures tend to be skewed, is there any effort to address it with specification tests and generalized linear models?

1. First, because the sample sizes were big difference between the treatment and comparison groups (584 vs. 2 608), using the nearest-neighbor method at a 1:4 ratio with 0.02 in the caliper ,. Second, we tried 1:1,1:2, 1:3,1:4, and 1:5 matching, we determined the optimal matching 1:4. 

2. Given that the skewed distribution of outpatient and inpatient expense violated the normal distribution assumption of the ordinary linear model, logarithmic transformation was adopted in the analysis.

Specific Comments

Line 140: Table 1: A t-test comparing the “treatment” and the “control” group would be useful.

Thanks for the reviewer's suggestion. We have revised Table 1 and made chi-square test for treatment group and comparison group.

Line 149: Table 2, “Not poor self-reported health” should be rephrased, maybe “self-reported good or better health”

Thanks for the reviewer's suggestion. We have revised Table 1 and Table 2, divided the self-reported health status into 0. Fair; 1. Poor; 2. Good; 3. Very good or Excellent, and reanalyzed the data

Line 22-23: it should be “in hypertension related inpatient expenditure”

Thanks for the reviewer's suggestion, we have made the revision as required here.

Line 169-173: need a concise title for Figure 4 – could put some of the information in a footnote.

Thanks for the reviewer's suggestion. Here we have changed it to: Fig 4. A. Trends in outpatient expenditure after PSM (RMB). B. Trends in inpatient expenditure after PSM (RMB), and the deleted content was condensed and placed in the main text.

Line 178: The authors occasionally used “compared to” – note that “compare to” highlights similarities while “compare with” stresses on the differences.

Thanks to the reviewer's suggestion, we have changed "compare to" to "compare with" in the full text.

Line 102-108: Sex should be (Men or women), “living with or without a partner” could be cohabitant – might need to be clarified or recoded. Education categories could be cleaned up as well. The income thresholds need to have a brief justification – it will be confusing to readers why those are used. “insurance” – should be health insurance. By the way, what are the insurance?

Thanks for the reviewer's suggestion. It has been revised as requested by the reviewer：

1.Sex (Men or Women); 

2.Marital status (Cohabitant or Single); 

3. Education is a simplified version of the 1997 International Standard Classification of Education (ISCED) codes. (www.uis.unesco.org): 1 = Less than lower secondary education, 2 = Upper secondary & vocational training, and 3 = Tertiary education; 

4. Household income was classified by four groups according to the 2011 household income quartile: poor (< RMB 2,320), low income (RMB 2,320–11,200), middle income (RMB 11,200–32,400), and high income (≥ RMB 32,400);

5. We have changed "insurance" to "health insurance". 

Participants who had any of the following insurance were defined as having health insurance: 1. Urban employee medical insurance; 2. Urban and rural resident medical insurance; 3. Urban resident medical insurance; 4. New rural cooperative medical insurance; 5. Government medical insurance; 6. Medical aid; 7. Urban non-employed persons's health insurance

---

## [Editor Report · Decision Letter 1]

9 Nov 2022

The impact of increasing expenditure on National Essential Public Health Services on the medical costs of hypertension in China: A difference-in-difference analysis

PONE-D-22-22961R1

Dear Dr. Long,

We’re pleased to inform you that your manuscript has been judged scientifically suitable for publication and will be formally accepted for publication once it meets all outstanding technical requirements.

Kind regards,

Jalandhar Pradhan

Academic Editor

PLOS ONE
---

## [Editor Report · Acceptance letter]

16 Nov 2022

PONE-D-22-22961R1 

The impact of increasing expenditure on National Essential Public Health Services on the medical costs of hypertension in China: A difference-in-difference analysis 

Dear Dr. Xue:

I'm pleased to inform you that your manuscript has been deemed suitable for publication in PLOS ONE. Congratulations! Your manuscript is now with our production department. 

Kind regards, 

on behalf of

Dr. Jalandhar Pradhan 

Academic Editor

PLOS ONE